# Effects of brief exposure to misinformation about e-cigarette harms on twitter: a randomised controlled experiment

Caroline Wright [1], Philippa Williams [1], Olga Elizarova [2], Jennifer Dahne,[3,4] Jiang Bian [5], Yunpeng Zhao,[5] Andy S L Tan [6,7]

CW and PW are joint first authors.

For numbered affiliations see end of article.

**Correspondence to**
Dr Caroline Wright;
caroline.wright@bristol.ac.uk

## ABSTRACT

**Objectives** To assess the effect of exposure to misinformation about e-cigarette harms found on Twitter on adult current smokers' intention to quit smoking cigarettes, intention to purchase e-cigarettes and perceived relative harm of e-cigarettes compared with regular cigarettes.

**Setting** An online randomised controlled experiment conducted in November 2019 among USA and UK current smokers.

**Participants** 2400 adult current smokers aged ≥18 years who were not current e-cigarette users recruited from an online panel. Participants' were randomised in a 1:1:1:1 ratio using a least-fill randomiser function.

**Interventions** Viewing 4 tweets in random order within one of four conditions: (1) e-cigarettes are just as or more harmful than smoking, (2) e-cigarettes are completely harmless, (3) e-cigarette harms are uncertain, and (4) a control condition of tweets about physical activity.

**Primary outcomes measures** Self-reported post-test intention to quit smoking cigarettes, intention to purchase e-cigarettes, and perceived relative harm of e-cigarettes compared with smoking.

**Results** Among US and UK participants, after controlling for baseline measures of the outcome, exposure to tweets that e-cigarettes are as or more harmful than smoking versus control was associated with lower post-test intention to purchase e-cigarettes (β=−0.339, 95% CI −0.487 to −0.191, p<0.001) and increased post-test perceived relative harm of e-cigarettes (β=0.341, 95% CI 0.273 to 0.410, p<0.001). Among US smokers, exposure to tweets that e-cigarettes are completely harmless was associated with higher post-test intention to purchase e-cigarettes (β=0.229, 95% CI 0.002 to 0.456, p=0.048) and lower post-test perceived relative harm of e-cigarettes (β=−0.154, 95% CI −0.258 to −0.050, p=0.004).

**Conclusions** US and UK adult current smokers may be deterred from considering using e-cigarettes after brief exposure to tweets that e-cigarettes were just as or more harmful than smoking. Conversely, US adult current smokers may be encouraged to use e-cigarettes after exposure to tweets that e-cigarettes are completely harmless. These findings suggest that misinformation about e-cigarette harms may influence some adult smokers' decisions to consider using e-cigarettes.

**Trial registration number** ISRCTN16082420.

### Strengths and limitations of this study

► This is the first study to explore the effect of exposure to misinformation about e-cigarette harms on Twitter, showing that after brief exposure to tweets that e-cigarettes are as or more harmful than smoking, current smokers may be deterred from using e-cigarettes (measured with intention to purchase e-cigarettes) as a harm reduction strategy. They are also more likely to wrongly believe that e-cigarettes are more harmful than regular cigarettes. We used a randomised controlled experimental design, which reduces the threat of potential confounding from observed and unobserved variables.

► We excluded visual content from our exposures and focused on Twitter: more research could be done to explore the impact of these factors.

► Our study sample did not fully represent the populations they were drawn from, which may mean our findings are not generalisable.

## INTRODUCTION

Although e-cigarette use is not completely harmless,[1 2] there is a general agreement that the short-term health risks are considerably lower than smoking regular cigarettes.[2] Despite this growing consensus, several recent studies show misperceptions about the relative harms of e-cigarettes among current smokers are increasing in both the USA and the UK.[3–5] Between 2014 and 2019, the percentage of current adult smokers in the UK who thought e-cigarettes were less harmful than cigarettes decreased from 45% to 34% and an even lower proportion of people believe so among smokers who were not using e-cigarettes.[5] Many smokers also do not think that complete replacement of cigarettes with e-cigarettes would lead to major health benefits.[2] The USA has a similar trend, with the percentage of adults perceiving e-cigarettes as less harmful than cigarettes declining from 29.3% to 25.8% between 2017 and 2018. Over the same period there was an increase from 1.8% to 4.4% of US adult smokers perceiving

e-cigarettes as much more harmful than cigarettes.[4] The increasing trends of misperceptions about the relative harms of e-cigarettes compared with regular cigarettes are important for public health because perceived harms of e-cigarettes are associated with smokers' willingness to use e-cigarettes[6] as a harm reduction strategy.

Misperceptions, defined as false or inaccurate beliefs of the individual,[7] of e-cigarette harms may be related to exposure to misinformation—information that is incorrect or misleading.[8] Based on the state of the science of e-cigarette harms,[1 2] misinformation related to e-cigarette harms was classified as the statements that either claim that e-cigarettes are equally or more harmful than smoking regular cigarettes or are completely harmless. As the evidence-base on e-cigarette harms has developed, related media and public discussion has involved uncertainty, defined as existing 'when details of the situation are ambiguous, complex, unpredictable, or probabilistic; when information is unavailable or inconsistent'.[9] Therefore, the impact of exposure to statements that claim the evidence of e-cigarette harms are uncertain are also important.

Health information is commonly accessed online, with 63% of UK adults using the internet to look for health-related information,[10] and 75% of US adults using the internet as their first source of health information.[11] People are increasingly encountering health information through social media platforms such as Twitter or Facebook.[12] These platforms enable users to generate and share content[13] and contrary to other media, there is often limited verification of the accuracy of health information.[14 15] A systematic review found user generated content was often inconsistent with clinical guidelines and health misinformation was increasingly available online.[16] We therefore focused on misinformation of e-cigarettes occurring on social media. We used Twitter data because they are free and publicly available and because of the documented prevalence of health misinformation on Twitter.[17 18] It is estimated that just over one in 5 Americans (22%) and 45% of social media users in the UK use Twitter.[19 20]

This study comprised US and UK participants as the contrasting policy approaches toward e-cigarette use across the two countries may mean that US and UK participants view harms associated with e-cigarettes differently. While the US approach focuses on protecting non-smokers from uptake of smoking via e-cigarette use, the UK's approach emphasises e-cigarettes as a harm reduction strategy to reduce the burden of risk on current smokers.[21] Further, the UK also has much stricter regulations relating to e-cigarette advertising and nicotine content of e-liquids compared with the US.[5]

To date, most studies have focused on health misinformation in relation to communicable diseases[8] and there is limited research on misinformation related to tobacco product use including e-cigarettes. While existing studies examined current perceptions of e-cigarette harms, little is known about the role of exposure to misinformation on

social media on these perceptions, and consequently on e-cigarette intentions and use.[22] To address this research gap, we conducted a web-based randomised controlled experiment to assess the effect of exposure to misinformation about e-cigarette harm found on Twitter, on smokers' intentions to quit smoking, intentions to purchase e-cigarettes, and perceptions of the relative harm of e-cigarettes compared with regular cigarettes.

## METHODS

### Study design

We used a randomised controlled experimental design.[23] The study was conducted using the online consumer research panel, Prodege which recruited participants from the US and the UK via internet sources (ie, email invitations, telephone alerts, banners and messaging on web sites and online communities). Participants' received reward points as per Prodege policies.

### Participants

Study participants were 2400 self-reported adult smokers aged ≥18 years, who were not currently using e-cigarettes. Informed consent was obtained electronically through the survey platform.

### Randomisation and masking

Following eligibility screening and having provided informed consent, participants completed baseline measures of study outcomes. Participants were then randomised to one of four experimental conditions: (1) e-cigarettes are as or more harmful than regular cigarettes, (2) e-cigarettes are completely harmless, (3) uncertain messages about e-cigarettes, and (4) messages for the control condition about physical activity from Twitter. Participants were randomised in a 1:1:1:1 ratio using the in-built least-fill randomiser function on the Prodege survey platform. Randomisation ensures that all participants have an equal chance of being assigned to each of the exposure conditions, and as such eliminates selection bias and associated problems with confounding. Adjusting for covariates is thus not needed in subsequent analysis, provided randomisation has been successful and covariates are equally distributed across experimental conditions.

### Procedures

Participants were told they would be shown different types of health-related information and asked for their opinions about e-cigarettes. Next, participants' provided baseline measures for: intention to quit smoking, intention to purchase e-cigarettes and perceived relative harm of e-cigarettes compared with regular cigarettes. After randomisation to a condition, they viewed one tweet at a time in random order (four tweets in total) and were asked brief questions about each tweet-perceived effectiveness of the tweet, likelihood of replying, retweeting, liking and sharing the tweet, and their emotional response

How about let's focus on the matter at hand which is Big Tobacco duping an entire generation - again - on our watch. Let's get the word out and educate young and old, vaping and E-cigarettes kill. @RAI_News

Smoking takes decades to cause cancer. Vaping, it seems, takes only a few years. The evidence is clear enough for US to ban flavoured vaping today, yep, today. The rest will follow as facts emerge, I imagine. It's pretty disgusting anyway -try it.

Vaping is still pretty much just as dangerous as cigs bc everything goes DIRECTLY into your lungs. Oh I forgot to mention the flavoring chemicals in vapes can also cause cancer. Seriously, look up actual medical research please.

#JUUL should be banned immediately. Anyone thinking vaping cigarettes is better than smoking is being conned. Vaping chemicals into your lungs will kill you. Altria is a murderer. Flat out mass murder. $MO

**Figure 1** Condition 1: e-cigarettes are as or more harmful.

to the tweet, more details of these questions can be found in online supplemental material 1. Next, they completed post-test measures of the study outcomes, current tobacco use behaviours, health information exposure, media use and sociodemographic and psychological characteristics. The average time taken to complete the survey was 29 min.

We captured tweets about e-cigarette harms using a validated machine learning algorithm the study team developed in an earlier phase of this research.[24 25] Using the random sample function within SPSS we selected a random 1% sample (n=499) of these tweets. Next, the study team narrowed this sample of tweets to 20 tweets per experimental condition using the following criteria: (1) explicit statement that e-cigarettes were either as or more harmful than smoking, completely harmless, or uncertain; (2) no mention of children or young people: (3) no mention of specific diseases; (4) no profanities; (5) had multiple 'likes' or 'retweets'; (6) no advertising; (7) no pictures; and (8) was available publicly (ie, not deleted).

We selected four representative tweets for each of the three experimental conditions: (1) e-cigarettes are as or more harmful, (2) e-cigarettes are completely harmless, and (3) uncertain message about e-cigarettes. Tweets for the control condition comprised four tweets about physical activity from Twitter. We selected physical activity promotion messages as the control condition to reduce potential bias due to experimenter demand and avoided topics related to e-cigarettes such as other forms of tobacco, alcohol or substance use behaviours. Figures 1–4 and online supplemental material 2 display

the content from the tweets that comprised each experimental condition.

## Outcome measures

### Baseline and post-test intention to quit smoking

Participants were asked to consider a smoking cessation contemplation ladder.[26] They were asked: "You have told us that you are currently smoking cigarettes. Each number below represents where various smokers are in their thinking about quitting. Please enter a number that indicates where you are now, ranging from "No thought of quitting" (0) to "Taking action to quit (eg, cutting down, enrolling in a program)" (10).

### Baseline and post-test intention to purchase e-cigarettes

Participants were asked: "How probable is it that you will purchase e-cigarettes in the next month?" Answer options ranged from "No chance, almost no chance" (0) to "Certain, practically certain" (10).[27]

### Baseline and post-test perceived relative harm of e-cigarettes compared with regular cigarettes

Participants were asked: "Compared to smoking cigarettes, would you say that electronic cigarettes are" Much less harmful (1) to much more harmful (5). This question included the option of don't know.[28] Two hundred and thirty-three participants answered 'don't know' to this question either at baseline or post-exposure and as

WOW. You're a doctor and you are spreading this fearmonger propaganda? What happened to your oath to do no harm?

There are ZERO proven harms in the 15 years vaping has existed when used in the suggested parameters.

I highly suggest you educate yourself on all the facts.. 1/2

I'm an asthmatic lol. I know the science behind vaping. It's completely safe. Big tobacco scares ppl. Like thruth dot .org... big tobacco supports them. It's crazy.

Oh, it's not only safer, they are SAFE - or, you know of any harm by vaping tho ~15y on the market and ~50.000.000 users world-wide? - and ~8.000 flavours! No, didn't think so bcos NONE so far - NONE - that'ts how SAFE vaping is - did say vaping. Any objections to that?

I don't worry about the ingreds of ejuice for #vaping, they are harmless, but I do wonder about the artificial breathing, the regular deep puffing. Do trumpet players get a breathing disorder? My puffing #ecigs is kind of like that.

**Figure 2** Condition 2: e-cigarettes are completely harmless.

We Still Don't Know How Safe #Vaping Is - it's time to get more information about the risks of #ecigarettes: @nytimes editorial

And people are like "but it's not that bad because it's not smoke" ok but nicotine is harmful with or without smoke and there is very limited research done on e-cigs so the FDA doesn't know how harmful they actually are to the extent that we know that cigarettes are harmful

This whole anti-vaping schtick is cooked up by drug regulation & enforcement to make sure the MONEY keeps flowing to their coffers.

I have yet to see a single credible piece of evidence showing that vaping causes real harm.

(As in more harm than drinking too much coffee.)

Is San Francisco's vaping ban backed by science?: San Francisco has decided to ban the sale of e-cigarettes in 2020, hoping to curb a surge in vaping among adolescents. But is the policy backed up by the available evidence? – How harmful is vaping? –...

**Figure 3** Condition 3: messages expressing uncertainty about e-cigarettes.

such were not included in the analysis. Participants who answered 'don't know' to the baseline question and post-exposure regarding relative harm distribute evenly across

Today reinforces my passion to push the need to exercise for not only the physical benefits. Get out and do something active for your mental health. Go for a walk and clear your mind. Find someone to join you & talk to them. My prayers go out to all today ❤

Adults (18+) need 150 minutes per week of moderate-intensity physical activity to improve and maintain health.

Let's #BeActive! 🏃♂️🧘♀️🏊♂️🚴♂️🤸♀️

It's #WorldMentalHealthDay 🎗 and we know sport and physical activity can have a powerful and positive effect on our mental wellbeing. That's why we invest in projects that are changing lives.

Physical activity & Exercise can have immediate and long-term health benefits. Most importantly, regular activity can improve our quality of life.

**Figure 4** Condition 4: messages about physical activity.

the experimental conditions and therefore pose no problem with respect to confounding or selection bias.

### Demographic and health information

Participants were asked to provide sociodemographic information including age, sex, race, ethnicity, highest education level, number of days of cigarette smoking in the past 30 days, ever use of e-cigarettes, information search about e-cigarettes, and social media use (see table 1).

### Statistical analysis

We used GPower (V.3.1)[29] to estimate effect sizes in the outcome variables as a function of message condition, assuming two-tailed tests, with 80% power and α=0.05. Based on these analyses, a final sample size of 2400 (600 in each arm) was deemed sufficient power to detect small effects in between-subject analyses of the main effect of condition among adult smokers (f=0.07). In stratified analyses by country, a sample size of 1200 (300 in each arm) will also ensure sufficient power to detect small effects between conditions (f=0.10).

Analyses were completed in 2020. Randomised controlled trials aim to compare groups of participants that differ only with respect to the intervention,[30] in this case exposure to misinformation. We performed univariate analyses for all study variables. Next, we analysed whether participants across conditions differed in terms of individual characteristics. To address the study aims, we used linear regression to predict post-test intentions to quit smoking, intentions to purchase e-cigarettes, and perceived relative harm of e-cigarettes by experimental condition compared with the control condition, adjusting for baseline measures of each outcome, respectively. Owing to overdispersion of the second outcome measure, intentions to purchase e-cigarettes, we additionally ran negative binomial regression models. We also ran sensitivity analyses, including country as a covariate (owing to the differences in baseline measurements between the USA and the UK; analysis using robust standard errors and bootstrapping—owing to non-normal distribution of residuals). We further conducted stratified analyses to compare the effects of experimental condition on each study outcome among US and UK participants separately. We also tested for interactions between experimental conditions and country (USA or UK). Stata V.15.1 was used to conduct all analyses.[31]

### RESULTS

Participants were 2400 adult current smokers recruited between 8 and 28 November 2019 (see figure 5: Consolidated Standards of Reporting Trials diagram). They were aged 18–84 years (mean=47.0, SD=14.58), 46.8% were female, 70.9% of the US participants were white, 16.8% black or African American and 12.3% were of other racial background, 90.3% of US participants were non-Hispanic. While 93.3% of the UK cohort were white and

**Table 1** Sociodemographic characteristics of study sample by experimental condition and country

| | USA | | | | UK | | | |
|---|---|---|---|---|---|---|---|---|
| Condition | 1: as or more | 2: completely harmless | 3: uncertainty | 4: control | 1: as or more | 2: completely harmless | 3: uncertainty | 4: control |
| Characteristics | n=300 | n=300 | n=300 | n=300 | n=300 | n=300 | n=300 | n=300 |
| **Age: Mean (SD)** | 50.5 (13.6) | 50.0 (13.6) | 50.0 (14.7) | 50.3 (13.5) | 44.1 (14.6) | 44.2 (14.4) | 44.0 (14.8) | 42.8 (14.6) |
| **Sex: No. (%)** | | | | | | | | |
| Female | 153 (51.0) | 154 (51.3) | 154 (51.3) | 140 (46.7) | 126 (42.0) | 136 (45.3) | 125 (41.7) | 135 (45.0) |
| **US Race: No. (%)** | | | | | | | | |
| White | 206 (68.7) | 214 (71.3) | 211 (70.3) | 220 (73.3) | | | | |
| Black or African American | 51 (17.0) | 47 (15.7) | 52 (17.3) | 51 (17.0) | | | | |
| Other races | 43 (14.3) | 39 (13.0) | 37 (12.3) | 29 (9.7) | | | | |
| **US Ethnicity: No (%)** | | | | | | | | |
| Non-Hispanic | 271 (90.3) | 269 (89.7) | 270 (90.0) | 274 (91.3) | | | | |
| Hispanic | 29 (8.7) | 31 (10.3) | 30 (10.0) | 26 (8.7) | | | | |
| **UK Ethnicity: No. (%)** | | | | | | | | |
| White | | | | | 284 (94.7) | 276 (92.0) | 278 (92.7) | 282 (94.0) |
| Other ethnicity | | | | | 16 (5.3) | 24 (8.0) | 22 (7.3) | 18 (6.0) |
| **Education: No. (%)** | | | | | | | | |
| High/Secondary school or below | 83 (27.7) | 99 (33.0) | 91 (30.3) | 89 (29.7) | 118 (39.3) | 126 (42.0) | 122 (40.7) | 129 (43.0) |
| Some college/ further education college | 111 (37.0) | 122 (40.7) | 123 (41.0) | 110 (36.7) | 110 (36.7) | 103 (34.3) | 105 (35.0) | 105 (35.0) |
| College/ University degree or higher | 106 (35.3) | 79 (26.3) | 86 (28.7) | 101 (33.7) | 72 (24.0) | 71 (23.7) | 73 (24.3) | 66 (22.0) |
| **Smoking status: Mean (SD)** | | | | | | | | |
| No. days smoked in last 30 days | 28.9 (4.2) | 27.8 (5.9) | 27.7 (5.9) | 28.1 (5.4) | 27.5 (6.3) | 27.4 (6.9) | 26.7 (7.7) | 27.1 (7.00) |
| **E-cigarette use: No. (%)** | | | | | | | | |
| Never used e-cigarettes | 145 (48.3) | 144 (48.0) | 152 (50.7) | 158 (52.7) | 138 (46.0) | 124 (41.3) | 152 (50.7) | 148 (49.3) |
| **Have you ever looked for e-cigarette information: No. (%)** | | | | | | | | |
| Yes | 75 (25.0) | 81 (27.0) | 58 (19.3) | 76 (25.3) | 72 (24.0) | 78 (26.0) | 74 (24.7) | 82 (27.3) |
| **Frequency of hearing e-cigarettes harmful: No. (%)** | | | | | | | | |
| Not at all | 22 (7.3) | 37 (12.3) | 38 (12.7) | 16 (5.3) | 42 (14.0) | 54 (18.0) | 64 (21.3) | 45 (15.0) |
| A little | 68 (22.7) | 78 (26.0) | 83 (27.7) | 67 (22.3) | 123 (41.0) | 113 (37.7) | 131 (43.7) | 122 (40.7) |
| Some | 105 (35.0) | 102 (34.0) | 103 (34.3) | 96 (32.0) | 81 (27.0) | 90 (30.0) | 69 (23.0) | 75 (25.0) |

Continued

**Table 1** Continued

| Condition | USA | | | | UK | | | |
|---|---|---|---|---|---|---|---|---|
| | 1: as or more | 2: completely harmless | 3: uncertainty | 4: control | 1: as or more | 2: completely harmless | 3: uncertainty | 4: control |
| Characteristics | n=300 | n=300 | n=300 | n=300 | n=300 | n=300 | n=300 | n=300 |
| A lot | 105 (35.0) | 83 (27.7) | 76 (25.3) | 121 (40.3) | 54 (18.0) | 43 (14.3) | 36 (12.0) | 58 (19.3) |
| Frequency of hearing e-cigarettes harmless: No. (%) | | | | | | | | |
| Not at all | 132 (44.0) | 107 (35.7) | 150 (50.0) | 137 (45.7) | 122 (40.7) | 97 (32.3) | 115 (38.3) | 145 (48.3) |
| A little | 86 (28.7) | 100 (33.3) | 76 (25.3) | 75 (25.0) | 97 (32.3) | 104 (34.7) | 110 (36.7) | 85 (28.3) |
| Some | 56 (18.7) | 61 (20.3) | 55 (18.3) | 61 (20.3) | 53 (17.7) | 68 (22.7) | 53 (17.7) | 58 (19.3) |
| A lot | 26 (8.7) | 32 (10.7) | 19 (6.3) | 27 (9.0) | 28 (9.3) | 31 (10.3) | 22 (7.3) | 12 (4.0) |
| Twitter use: | | | | | | | | |
| Several times a day | 21 (7.0) | 29 (9.7) | 21 (7.0) | 29 (9.7) | 31 (10.3) | 31 (10.3) | 39 (13.0) | 32 (10.7) |
| About once a day | 16 (5.3) | 15 (5.0) | 32 (10.7) | 23 (7.7) | 22 (7.3) | 32 (10.7) | 33 (11.0) | 24 (8.0) |
| A few times a week | 28 (9.3) | 31 (10.3) | 25 (8.3) | 22 (7.3) | 28 (9.3) | 32 (10.7) | 24 (8.0) | 27 (9.0) |
| Every few weeks | 12 (4.0) | 8 (2.7) | 10 (3.3) | 17 (5.7) | 20 (6.7) | 17 (5.7) | 20 (6.7) | 17 (5.7) |
| Once a month or less | 22 (73) | 28 (9.3) | 27 (9.0) | 33 (11.0) | 21 (7.0) | 33 (11.0) | 21 (7.0) | 21 (7.0) |
| Never | 201 (67.0) | 189 (63.0) | 185 (61.7) | 176 (58.7) | 178 (59.3) | 155 (51.7) | 163 (54.3) | 179 (59.7) |

Test for variance across conditions; continuous variables analysed using one-way analysis of variance test, categorical variables analysed using $\chi^2$ test.

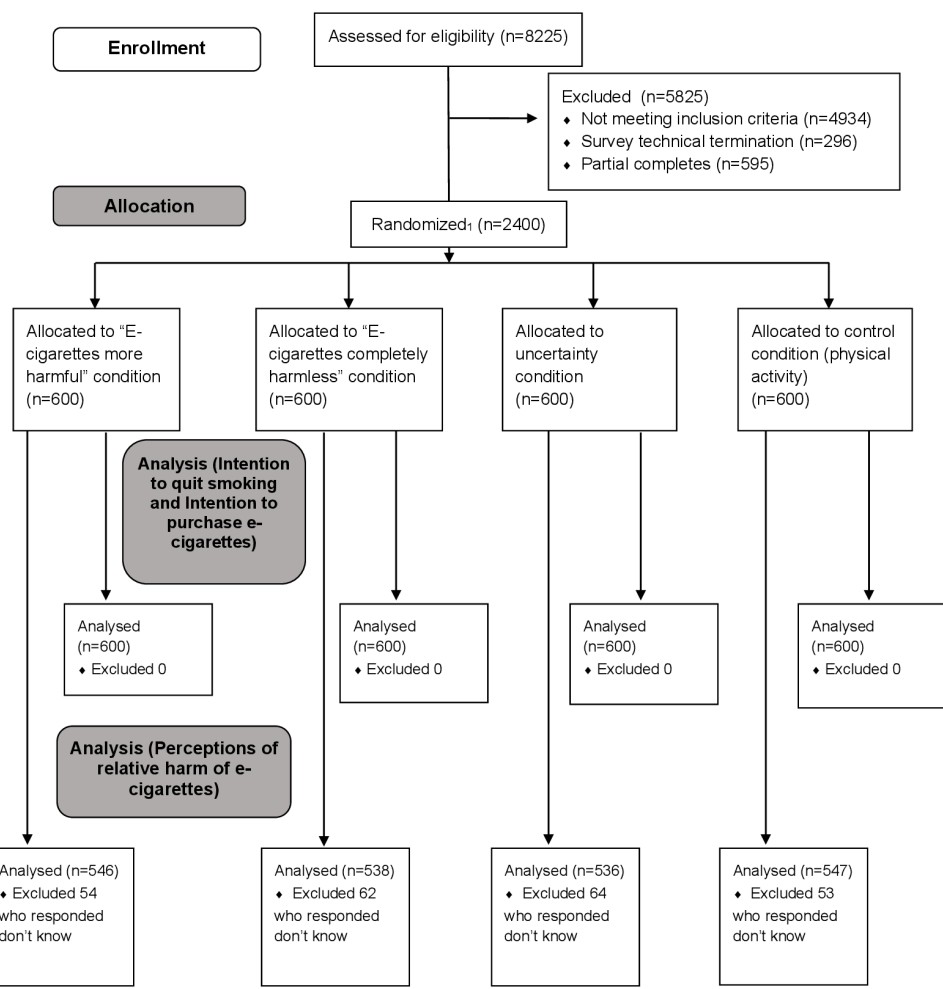

**Figure 5** Consolidated Standards of Reporting Trials flow diagram. Survey recruitment used a least-fill approach; as a respondent came in, they were assigned to the exposure with the lowest complete count.

6.7% were from other ethnic backgrounds. Most of the cohort (82.1%) smoked cigarettes every day and 51.6% had ever used e-cigarettes. Table 1 summarises the sample characteristics by experimental condition. We found that randomisation had been achieved and all covariates were distributed evenly across the four study conditions.

Three quarters of participants (n=1804, 75.2%) had not previously searched for information on e-cigarettes. Participants were more likely to report that they had heard that e-cigarettes are more harmful than cigarettes (n=1297, 54.0%), than hearing that e-cigarettes are harmless (n=662, 27.6%). Over half of the sample (n=1426, 59.4%) had never used Twitter, with Facebook being the most common social media platform used several times a day (n=1194, 49.8%).

At baseline, 25.2% of participants (n=605) placed themselves in the middle of the intention to quit ladder (mean=5.0, SD=3.0); this was similar for both US and UK participants. Over half the participants (n=1312, 54.7%) said that there was no chance/almost no chance that they would buy e-cigarettes in the next month. The distributions for intentions to buy were also very similar for US and UK participants. However, there were marked differences between the two populations with respect to

perceptions of the relative harm of e-cigarettes: nearly twice as many UK participants said that e-cigarettes are much less harmful than regular cigarettes compared with US participants. Similarly, more than twice as many UK participants said that e-cigarettes are less harmful than regular cigarettes (n=448, 37.3%), compared with US participants (n=222, 18.5%). Conversely, more than three times as many US participants thought that e-cigarettes are much more harmful than regular cigarettes (US: n=217, 18.1%, UK: n=69, 5.8%) and more than twice as many saw them as more harmful (US: n=128, 10.7%, UK: n=62, 5.2%).

We additionally compared the mean and SD for the outcome measures, both pre-exposure and post-exposure across the four conditions for the US and UK separately (table 2). We found that pre-exposure intentions to quit and perceptions of the relative harm of e-cigarettes were generally higher and intentions to purchase e-cigarettes were generally lower among US participants.

Tables 3 and 4 summarise the results from the regression analyses. The adjusted analysis includes both the experimental condition as the exposure and the baseline measure of the outcomes. We present the adjusted analysis here. Compared with the control condition, there was

**Table 2** Outcome measures by experimental condition and country

| Condition | USA | | | | UK | | | |
|---|---|---|---|---|---|---|---|---|
| | 1 | 2 | 3 | 4 | 1 | 2 | 3 | 4 |
| **Outcome measures** | | | | | | | | |
| **Intentions to quit smoking** | n=300 | n=300 | n=300 | n=300 | n=300 | n=300 | n=300 | n=300 |
| Pre-exposure: mean (SD) | 5.16 (2.94) | 5.25 (3.17) | 5.23 (3.00) | 5.48 (3.14) | 4.72 (2.85) | 4.73 (2.86) | 4.83 (2.90) | 4.78 (3.04) |
| Post-exposure: mean (SD) | 5.29 (2.96) | 5.34 (3.15) | 5.46 (3.04) | 5.72 (3.20) | 4.93 (2.90) | 4.80 (2.91) | 4.96 (2.89) | 4.93 (3.09) |
| **Intentions to purchase e-cigarettes** | n=300 | n=300 | n=300 | n=300 | n=300 | n=300 | n=300 | n=300 |
| Pre-exposure: mean (SD) | 1.33 (2.24) | 1.15 (2.08) | 1.25 (2.20) | 1.29 (2.23) | 1.67 (2.37) | 1.57 (2.33) | 1.88 (2.54) | 1.71 (2.47) |
| Post-exposure: mean (SD) | 0.98 (2.02) | 1.30 (2.27) | 1.16 (2.17) | 1.27 (2.31) | 1.21 (2.16) | 1.68 (2.56) | 1.73 (2.50) | 1.79 (2.61) |
| **Perceptions of relative harms of e-cigarettes** | n=274 | n=268 | n=274 | n=276 | n=272 | n=270 | n=262 | n=271 |
| Pre-exposure: mean (SD) | 3.17 (1.03) | 3.35 (1.28) | 3.20 (1.04) | 3.26 (1.10) | 2.64 (0.95) | 2.67 (0.93) | 2.60 (0.90) | 2.68 (0.90) |
| Post-exposure: mean (SD) | 3.45 (1.06) | 3.15 (1.12) | 3.22 (1.02) | 3.22 (1.07) | 3.02 (1.00) | 2.60 (0.98) | 2.60 (0.93) | 2.66 (0.92) |

no difference in the post-test intention to quit smoking among those who viewed tweets stating that e-cigarettes are as or more harmful than cigarettes, the completely harmless condition or tweets that are uncertain. The results did not change substantially in the stratified analysis (table 4).

Compared with participants assigned to the control group, there was a statistically significant reduction in post-test intention to purchase e-cigarettes for those exposed to the as or more harmful messages ($\beta$=−0.339, 95% CI −0.487 to −0.191, p<0.001). In the stratified analysis, the effect of viewing as or more harmful tweets on reducing intentions to purchase e-cigarettes was observed in both US ($\beta$=−0.312, 95% CI −0.522 to −0.073, p=0.011) and UK samples ($\beta$=−0.365, 95% CI −0.551 to −0.178, p<0.001). Further, the effect of viewing tweets that e-cigarettes are completely harmless was associated with an increase in intention to purchase e-cigarettes but

Compared with participants assigned to the control messages, participants who viewed the as or more harmful messages were significantly more likely to perceive e-cigarettes as *more* harmful than regular cigarettes ($\beta$=0.341, 95% CI 0.273, 0.410, p<0.001). Participants assigned to the completely harmless messages were significantly more likely to perceive e-cigarettes as *less* harmful than regular cigarettes ($\beta$=−0.106, 95% CI −0.174 to −0.037, p=0.003). These effects remained following stratification by country (UK: $\beta$=0.385, 95% CI 0.298, 0.476, p<0.001; US: $\beta$=0.296, 95% CI 0.193, 0.400, p<0.001). The effect of the completely harmless misinformation on participants perceiving e-cigarettes as less harmful than cigarettes was limited to the US population after stratification ($\beta$=−0.154, 95% CI −0.258 to −0.050, p=0.004).

We additionally ran a number of sensitivity analyses owing to differences in baseline measurement between the USA and the UK, and non-normality of residuals in the regression analyses. However, there were no substantial differences to report from any of the sensitivity analyses (see table 5). We additionally tested for interactions between experimental conditions and country (USA or UK), but found no evidence of an effect. A summary of the results is available through online supplemental video 1.

## DISCUSSION

Our results suggest that exposure to misinformation about e-cigarette harms influences adult smokers' decisions to purchase e-cigarettes and their perceived relative harm of e-cigarettes, compared with regular cigarettes. To our knowledge, this is the first study to test the effect of brief exposure to misinformation and uncertainty about e-cigarette harms found on Twitter on smokers' intentions to quit smoking, intentions to use e-cigarettes and perceptions of relative harm. Both US and UK samples of adult smokers were adversely affected by misinformation about e-cigarettes. We also observed that US smokers who viewed tweets that e-cigarettes were completely harmless

**Table 3** Regression analysis predicting intention to quit regular cigarettes, intention to purchase an e-cigarette and perceived relative harm of e-cigarettes compared with regular cigarettes (adjusted for baseline measures of outcome)

| | Intention to quit smoking regular cigarettes (n=2400) | | | Intention to purchase e-cigarette* (n=2400) | | | Perceived relative harm of e-cigarettes compared with regular cigarettes (n=2167) | | |
|---|---|---|---|---|---|---|---|---|---|
| | β | 95% CI | P value | β | 95% CI | P value | β | 95% CI | P value |
| Control (referent) | | | | | | | | | |
| As or more harmful | −0.031 | (−0.152 to 0.091) | 0.622 | −0.339 | (−0.487 to −0.191) | ≤0.001 | 0.341 | (0.273 to 0.410) | ≤0.001 |
| Completely harmless | −0.120 | (−0.241 to 0.002) | 0.054 | 0.111 | (−0.029 to 0.250) | 0.120 | −0.106 | (−0.174 to −0.037) | 0.003 |
| Uncertainty | −0.017 | (−0.139 to 0.104) | 0.780 | −0.106 | (−0.247 to 0.036) | 0.143 | −0.018 | (−0.051 to 0.086) | 0.615 |
| Pre-exposure intention to quit | 0.945 | (0.931 to 0.960) | ≤0.001 | – | – | – | – | – | – |
| Pre-exposure intention to purchase | – | – | – | 0.437 | (0.417 to 0.458) | ≤0.001 | – | – | – |
| Pre-exposure perceived relative harm of e-cigarettes | – | – | – | – | – | – | 0.841 | (0.818 to 0.864) | ≤0.001 |
| | | $R^2$=0.874 | | | Pseudo $R^2$=0.2125 Alpha=0: $p$≤0.001 | | | $R^2$=0.704 | |

*For intention to purchase e-cigarettes, negative binomial regression was conducted due to a zero-inflated distribution/non-normal distribution.

**Table 4** Regression analysis predicting intention to quit regular cigarettes, intention to purchase an e-cigarette and perceived relative harm of e-cigarettes compared with regular cigarettes stratified by country of residence status (adjusted for baseline measures of outcome)

| | Intention to quit smoking regular cigarettes (USA, n=1200; UK n=1200) | | | Intention to purchase e-cigarette* (USA, n=1200; UK n=1200) | | | Perceived relative harm of e-cigarettes compared with regular cigarettes (USA, n=1092; UK n=1075) | | |
|---|---|---|---|---|---|---|---|---|---|
| | β | 95% CI | P value | β | 95% CI | P value | β | 95% CI | P value |
| **USA** | | | | | | | | | |
| Control (referent) | | | | | | | | | |
| As or more harmful | −0.126 | (−0.305 to 0.054) | 0.169 | −0.312 | (−0.552 to −0.073) | 0.011 | 0.296 | (0.193 to 0.400) | ≤0.001 |
| Completely harmless | −0.161 | (−0.340 to 0.019) | 0.079 | 0.229 | (0.002 to 0.456) | 0.048 | −0.154 | (−0.258 to −0.050) | 0.004 |
| Uncertainty | −0.025 | (−0.204 to 0.155) | 0.786 | −0.102 | (−0.334 to 0.130) | 0.389 | 0.036 | (−0.067 to 0.140) | 0.492 |
| Pre-exposure intention to quit | 0.940 | (0.920 to 0.961) | ≤0.001 | – | – | – | – | – | – |
| Pre-exposure intention to purchase | – | – | – | 0.475 | (0.439 to 0.510) | ≤0.001 | – | – | – |
| Pre-exposure perceived relative harm of e-cigarettes | – | – | – | – | – | – | 0.807 | (0.773 to 0.841) | ≤0.001 |
| | | $R^2$=0.869 | | | Pseudo $R^2$=0.205 Alpha=0: p≤0.001 | | | $R^2$=0.666 | |
| **UK** | | | | | | | | | |
| Control (referent) | | | | | | | | | |
| As or more harmful | 0.063 | (−0.101 to 0.228) | 0.451 | −0.365 | (−0.551 to −0.178) | ≤0.001 | 0.385 | (0.297 to 0.474) | ≤0.001 |
| Completely harmless | −0.079 | (−0.244 to 0.085) | 0.344 | 0.034 | (−0.141 to 0.208) | 0.707 | −0.053 | (−0.142 to 0.035) | 0.238 |
| Uncertainty | −0.011 | (−0.176 to 0.154) | 0.895 | −0.113 | (−0.289 to 0.062) | 0.205 | −0.002 | (−0.092 to 0.087) | 0.958 |
| Pre-exposure intention to quit | 0.948 | (0.928 to 0.968) | ≤0.001 | – | – | – | – | – | – |
| Pre-exposure intention to purchase | – | – | – | 0.406 | (0.381 to 0.431) | ≤0.001 | – | – | – |
| Pre-exposure perceived relative harm of e-cigarettes | – | – | – | – | – | – | 0.875 | (0.840 to 0.909) | ≤0.001 |
| | | $R^2$=0.879 | | | Pseudo $R^2$=0.217 Alpha=0: p≤0.001 | | | $R^2$=0.701 | |

Above models controlled for pre-exposure measure of outcome.

*For intention to purchase e-cigarettes, negative binomial regression was conducted due to a zero-inflated distribution/non-normal distribution only after stratification (β=0.229, 95% CI 0.002, 0.456, p=0.048) and only among US participants.

**Table 5** Sensitivity analyses: adjusted regression analysis predicting intention to quit regular cigarettes, intention to purchase an e-cigarette and perceived relative harm of e-cigarettes compared with regular cigarettes—A: includes country as a covariate, B: analysis with robust standard errors and C: analysis with bootstrapping

| | Intention to quit smoking regular cigarettes (n=2400) | | | Intention to purchase e-cigarette (n=2400) | | | Perceived relative harm of e-cigarettes (n=2167) | | |
|---|---|---|---|---|---|---|---|---|---|
| | β | 95% CI | P value | β | 95% CI | P value | β | 95% CI | P value |
| **A** | | | | | | | | | |
| Control (referent) | | | | | | | | | |
| As or more harmful | −0.031 | (−0.153 to 0.091) | 0.620 | −0.337 | (−0.485 to −0.189) | ≤0.001 | 0.341 | (0.273 to 0.410) | ≤0.001 |
| Completely harmless | −0.120 | (−0.241 to 0.002) | 0.054 | 0.111 | (−0.029 to 0.250) | 0.120 | −0.105 | (−0.174 to −0.037) | 0.003 |
| Uncertainty | −0.017 | (−0.139 to 0.104) | 0.779 | −0.106 | (−0.247 to 0.035) | 0.142 | −0.017 | (−0.052 to 0.086) | 0.628 |
| **B** | | | | | | | | | |
| Control (referent) | | | | | | | | | |
| As or more harmful | −0.031 | (−0.156 to 0.095) | 0.633 | −0.339 | (−0.499 to −0.179) | ≤0.001 | 0.341 | (0.271 to 0.412) | ≤0.001 |
| Completely harmless | −0.120 | (−0.241 to 0.002) | 0.054 | 0.111 | (−0.036 to 0.258) | 0.141 | −0.106 | (−0.163 to −0.048) | ≤0.001 |
| Uncertainty | −0.017 | (−0.132 to 0.097) | 0.767 | −0.106 | (−0.253 to 0.042) | 0.160 | 0.018 | (−0.044 to 0.079) | 0.572 |
| **C** | | | | | | | | | |
| Control (referent) | | | | | | | | | |
| As or more harmful | −0.031 | (−0.147 to 0.085) | 0.605 | −0.339 | (−0.493 to −0.185) | ≤0.001 | 0.341 | (0.280 to 0.403) | ≤0.001 |
| Completely harmless | −0.120 | (−0.237 to 0.002) | 0.047 | 0.111 | (−0.040 to 0.262) | 0.151 | −0.106 | (−0.168 to −0.044) | 0.001 |
| Uncertainty | −0.017 | (−0.151 to 0.116) | 0.799 | −0.106 | (−0.239 to 0.028) | 0.121 | 0.018 | (−0.051 to 0.087) | 0.617 |

reported lower perceived harms of vaping and higher intentions to purchase e-cigarettes in this study. This effect was absent among UK smokers. This difference between US and UK smokers may be due to the differing policy contexts of the countries. However, further research is needed to assess underlying policy and contextual factors that explain these differences between countries in the effects of e-cigarette misinformation.

These findings are important because they show that after brief exposure to tweets that e-cigarettes are as or more harmful than smoking, current smokers may be deterred from using e-cigarettes (measured with intention to purchase e-cigarettes) as a harm reduction strategy. They are also more likely to wrongly believe that e-cigarettes are more harmful than regular cigarettes. However, more research is needed to assess whether misinformation exposure about e-cigarette harms will negatively influence smokers' behaviours to reduce harms from using combusted cigarettes by opting for less harmful forms of nicotine delivery using e-cigarettes. There is consensus that debunking or correcting exposure to misinformation is extremely challenging, common techniques have even been found to further engrain misinformation.[15 32] Reducing exposure to misinformation has its own challenges, as misinformation on social media spreads more pervasively than accurate information and the spread is due to mostly human actions, rather than automated bots.[33] In addition, it is often hard to categorise content as misinformation, especially when the evidence around a given health topic is inconclusive, or the way the information is communicated is unclear. This creates challenges in both harnessing algorithms to alert users to misinformation and also communicating ways to spot misinformation. These points, combined with our findings, have the potential to undermine the efforts of the public health community to reduce harm among current smokers. However, innovative health communication approaches need to be developed and tested to both reduce exposure to and counter misinformation using effective harm reduction and health promotion strategies. Strategies are already being employed by social media platforms to address the problem of misinformation, for example, downranking content and removing or blocking users with content identified as misinformation. While it may be difficult to keep up with and identify health misinformation as such, it is possible to warn smokers of the problem of misinformation and encourage them to seek out their health information from official sources. Healthcare providers' should be aware that their patients may have seen misinformation on social media and hold incorrect beliefs about e-cigarettes. They should always correct these and consider the ways they can help their patients to identify accurate health information. Finally, governments and policy-makers should make sure all social media searches for e-cigarettes are flagged with official health guidance. They should also regulate all forms of misinformation on social media and improve people's awareness and ability to find accurate information.

There are several limitations of the study, first, we excluded visual content from the exposures to ensure that the format of tweets was consistent across conditions and participants were focused on the content of the tweets. However, prior studies indicate that visual cues within e-cigarette advertisements are associated with perceptions about and decisions to use e-cigarettes.[8 34] Second, health misinformation is spread in different ways. We used Twitter data because they are free and publicly available and because of the documented prevalence of health misinformation on Twitter.[17 18] However, over half of participants (59%) indicated they did not use Twitter meaning they may not be familiar with viewing or engaging with tweets. To address this, we included definitions of each of these engagement behaviours, prior to responding to questions on the likelihood of replying, retweeting, liking or sharing each message. Further, our findings are still useful because intentions are strong predictors of behaviour, as shown by Ajzen's Theory of Planned Behavior.[35] Misinformation is ubiquitous—Allcott et al found a total of 672 sites producing false stories or unique fake news sites.[36] Stories from these sites are shared on Facebook, Twitter and cross-posted on other social media platforms. Therefore, while this sample may not be exposed to misinformation on Twitter in real life, they are likely exposed via different channels. Third, there is the issue of the reliability of self-reported smoking compared with biochemical verification of smoking status. However, given that we used an online self-administered survey, it is unlikely to have a big impact on participants' answers. Further, it's been shown that self-reported smoking prevalence, checked by biochemical verification, was underestimated by only 0.6% in the USA and 2.8% in the UK.[37] Fourth, our study sample was not fully representative of the populations they were drawn from. For example, white people make up 86% of the UK population, but represented 93.3% of the UK sample in this study, which may mean our findings are less generalisable. Fifth, previous research on health misinformation on social media identified important factors that might play a role in the mechanism of action of misinformation. Among those factors are the type of content, the source of the message, the sender's authoritativeness, the argument length, the novelty, timing, repetition and hashtags. We were not able to examine the impact of these message features in detail. Future research is needed to determine the effects of varying these features on smokers' processing of misinformation about e-cigarettes.[33 38 39] Finally, there was an outbreak of e-cigarette or vaping product use-associated lung injuries that were first identified in August 2019 in the US and subsequently traced to products containing tetrahydrocannabinol from the illicit market. This outbreak, in combination with the different contexts of the two countries, may have influenced participants' views on e-cigarettes during the time of the study data collection. However, because of the experimental design to randomly assign participants into conditions, we do

not anticipate that this would have biased our findings systematically.

Future research should focus on identifying the factors that make misinformation effective and how it is perceived by exposed individuals. Conducting research using different social media platforms, study designs and analytical tools, and focusing on analysing the message or communication factors are all important. According to our study, Facebook was overwhelmingly the social media platform used by these participants. It would therefore be interesting to replicate this research using Facebook. Second, there is a need to explore the role of cognitive factors, beliefs, past experiences and other individual level factors in the effects of misinformation. For instance, based on the theory of bias assimilation stating that people gravitate to information they have previously heard, future research should test whether the observed results could be explained by the fact that many individuals were previously exposed to misinformation. Third, it is important to refine and further develop a reliable algorithm that could distinguish between accurate and misinformation about e-cigarettes. With the amount of information that is currently generated by users on different social media platforms, an automated approach of identifying misinformation could be most cost-effective and timely. Nevertheless, any algorithms, evident from our prior work,[24][25] cannot achieve 100% accuracy, leading to misclassification errors and require constant refinement and evaluation as new types of misinformation emerge. Fourth, we were not able to examine the impact of specific features of the tweets, for example the source of the message or the sender's authoritativeness. Future research is needed to determine the effects of varying these features on smokers' processing of misinformation about e-cigarettes. Our exposure was only brief; therefore, future research to evaluate the effect of longer or repeated exposures to misinformation would also be useful, to assess the effects on e-cigarette use intentions and subsequent vaping or smoking behaviours. Finally, future research could extend our analysis to include behaviours as well as intentions.

## CONCLUSIONS

US and UK adult current smokers may be deterred from considering using e-cigarettes after brief exposure to tweets that e-cigarettes are as or more harmful than smoking. Conversely, US adult current smokers may be encouraged to use e-cigarettes and view them as less harmful than regular cigarettes, after exposure to tweets that e-cigarettes are completely harmless. These findings suggest that misinformation about e-cigarette harms may influence adult smokers' decisions to consider using e-cigarettes.

**Author affiliations**
[1]Population Health Sciences, Bristol Medical School, University of Bristol, Bristol, UK
[2]Play Collaborate Change, Boston, Massachusetts, USA
[3]Psychiatry and Behavioral Sciences, Medical University of South Carolina College of Medicine, Charleston, South Carolina, USA
[4]Hollings Cancer Center, Medical University of South Carolina, Charleston, South Carolina, USA
[5]Health Outcomes & Biomedical Informatics, College of Medicine, University of Florida, Gainesville, Florida, USA
[6]Annenberg School for Communication, University of Pennsylvania, Philadelphia, Pennsylvania, USA
[7]Leonard Davis Institute of Health Economics, University of Pennsylvania, Philadelphia, Pennsylvania, USA

**Acknowledgements** We would like to thank Cancer Research UK who funded this research.

**Contributors** CW, ASLT, OE, JD and JB contributed to the original research idea. JB and YZ did the machine learning and JB and PW annotated the tweets. All authors contributed to the design of the survey instrument. CW and PW did the statistical analysis with input from all the authors. All authors contributed to the drafting and editing of the paper.

**Funding** Research reported in this publication was supported by a Cancer Policy Research Centre Innovation grant (C60153/A28664). CW is funded by a Cancer Research UK Population Research Postdoctoral Fellowship (C60153/A23895). JD is supported by the National Institute on Drug Abuse, K23 DA045766. We would also like to thank the National Cancer Institute (NCI) for their support during the sandpit event where this research idea came about.

**Competing interests** None declared.

**Patient consent for publication** Not required.

**Ethics approval** The University of Bristol's Institutional Review Board, Faculty of Health Sciences Research Ethics Committee (FREC) approved this study. Ref: 80323 The tweets used in this research are all in the public domain and participants could therefore have been exposed to this misinformation at any time. We further provided participants with a debrief of accurate information about e-cigarettes compared with regular cigarette harms as well as information about smoking cessation services.

**Provenance and peer review** Not commissioned; externally peer reviewed.

**Data availability statement** Data are available upon reasonable request. The data are deidentified participant data as outlined at http://www.isrctn.com/ISRCTN16082420. The data will be made available upon reasonable requests. We also plan to publish the data from the study once all research outlined in our research proposal has been submitted for publication.

**ORCID iDs**
Caroline Wright http://orcid.org/0000-0002-4321-4872
Philippa Williams http://orcid.org/0000-0002-6774-2514
Olga Elizarova http://orcid.org/0000-0001-5239-4759
Jiang Bian http://orcid.org/0000-0002-2238-5429
Andy S L Tan http://orcid.org/0000-0001-6459-6171

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
