## [Reviewer comments · BMJ Open]

ARTICLE DETAILS

TITLE (PROVISIONAL)	Effects of Brief Exposure to Misinformation about E-cigarette Harms on Twitter: A Randomised Controlled Experiment
AUTHORS	Wright, Caroline; Williams, Philippa; Elizarova, Olga; Dahne, Jennifer; Bian, Jiang; Zhao, Yunpeng; Tan, Andy

VERSION 1 – REVIEW

REVIEWER	Nguyen, Nhung UCSF
REVIEW RETURNED	17-Dec-2020

GENERAL COMMENTS	Overall, the current study aimed to examine the effect of brief exposure to misinformation and uncertainty about e-cigarette harms on Twitter on smokers' intentions to quit smoking, intentions to use e-cigarettes, and perceptions of relative harm. The study found that misinformation about e-cigarette harms may influence some adult smokers' decisions to consider using e-cigarettes. The manuscript is well-written. However, there are several unclear points in the Methods needed to clarify, as below. 1. A justification for choosing Twitter (but not other social media sites) as a web-based platform for the experimental should be provided in the Introduction (although it mentioned briefly in the Limitation). If available, provide national percentages of Twitter users in US and UK.2. An important point which is not clear is that how to know that participants had been not exposed to the experimented tweets before the study. If they already had seen the assigned tweets before, measures of outcomes may not be the same if they hadn't.3. Methods - "Randomisation and Masking": It said that "Participants were then randomised to one of four experimental conditions...", however, only 3 conditions were listed. In addition, the statement "Adjusting for covariates is thus not needed in subsequent analysis" maybe not correct since adjusting for covariates may be still needed in subsequent analysis if distributions of confounders are not equal across experimental conditions.4. Removing 233 participants who answered 'don't know' to the question on baseline and post-test perceived relative harm of e-cigarettes compared to regular cigarettes out of the analysis may introduce bias if these participants didn't distribute equally across the experimental conditions. I suggest to include this answer option in the analysis.5. Tables 1 and 2 need to provide p-values for testing differences
--

	across 4 conditions for the US sample and the UK sample. In addition, name of each condition should be provided rather than the numbers. 6. Tables 3 and 4 can be combined into 1 table 7. Why the variable “country” was not included in the main analysis, but only in the sensitive analysis? Was the interaction between “country” and “experimental conditions” examined? 8. What were the purpose of the sensitivity analyses with robust standard errors and bootstrapping? 9. How repeated measures within participants were accounted for in the current analysis? 10. How regression models were adjusted for baseline measures of each outcome? Maybe add these baseline variables in tables for clarification.
--	--

REVIEWER	Holmes, Louisa The Pennsylvania State University Department of Geography, GEOG
REVIEW RETURNED	23-Dec-2020

GENERAL COMMENTS	This study evaluates the impacts of “misinformation” shared via social media on the likelihood of e-cigarette uptake, and perceptions of e-cigarette harm, among an online panel of current cigarette smokers who have never used e-cigarettes in the USA and the UK. The study is sufficiently powered and the comparison regarding e-cigarette and cigarette cessation intent between the USA and the UK, two countries with substantially different policy approaches to e-cigarettes, is innovative. However, the manuscript has some flaws that should be addressed prior to publication. ABSTRACT  Line 30, fix so it reads “as or more” rather than “as more” INTRODUCTION  This manuscript assumes definitions of “misperception” and “misinformation” without ever indicating how those are defined for purposes of the study, or whether they might be subject to bias on the part of the authors (who seem to prefer a harm reduction approach); there are also no citations to support these assumptions so it is difficult to discern whether “misperception” is conflated with “unsupportive of harm reduction.” METHODS  In the Methods and Discussion, the authors appear to equate randomization to the study conditions with population distribution, which are not the same (p8, lines 45-50; p13, lines 19-22) – given that recruitment did not reflect population distribution by demographic characteristics, and that e-cigarette use varies substantially along these characteristics (e.g. race/ethnicity, age, socioeconomic position), these items need to be controlled, reported and evaluated in the analyses P9, lines 5-7: what are the questions that were asked regarding each tweet?
---

	 • What was the time frame of evaluation from pre- to post-test? • P9, lines 20-23: how did the team narrow from 499 tweets to 20? What types of criteria were used? • Authors indicate that 50% of the study population did not use Twitter, which is problematic given the study approach; since only half of the population is likely to see Tweets about e-cigarettes, there needs to be more discussion of why this doesn't skew the study results substantially • Literature has demonstrated demographic differences, on average, between Twitter and Facebook users, as well as between those who use social media and those who don't, so authors need to either adjust for this in their models, or list it as a serious limitation (see for example: https://doi.org/10.1177%2F2053168017720008) • There is no discussion of who the authors of the included Tweets are and how this might impact the way participants (or social media users more generally) interpret the information tweeted, e.g. in Figure 1, the author of the first tweet identifies herself as the founder of an organization, which may confer more authority on her in the participants' mind, as might the fact that Ross Gerber has a blue check, compared with "Brea"  o Additionally, there has been much discussion in media, including with Jack Dorsey, about how more likes tends to mean Twitter users are more likely to pay attention to certain Tweets, and there is clear difference in these figures in the number of likes • In Figure 3, the tweet by "Mom folding laundry" is clearly disputing the idea that e-cigarettes are harmful; how can this be classified as "messages expressing uncertainty?" RESULTS  • Given the differences in smoking behavior, intent to quit, and e-cig use behavior by sociodemographic characteristic and nationality, it's imperative that regression results for demographic covariates be included • Table 1: does White include Non-Hispanic White? In the USA, Latinos have been shown to have substantially different tobacco use patterns than Whites account for nearly 20% of the total population LIMITATIONS  • I would suggest that three additional limitations of the study that deserve attention are:  o Lack of inclusion of cannabis use as a covariate given its legality in many parts of the USA and the popularity of using e-cigarettes to vape it o Lack of inclusion of other tobacco products, or polytobacco use, which may confound intentions about e-cigarette use / opinions on e-cig harm o Intention has not been demonstratively linked to behavior among tobacco users DISCUSSION  • I would respectfully suggest that the differences in results by country are one of the more interesting aspects of the study and that these should be included in the discussion with attention to how cultural and policy approaches to e-cigarettes clearly seem to have an impact on intentions and harm perception
--	--

VERSION 1 – AUTHOR RESPONSE

Reviewer: 1

Dr. Nhung Nguyen, UCSF

Comments to the Author:

Overall, the current study aimed to examine the effect of brief exposure to misinformation and uncertainty about e-cigarette harms on Twitter on smokers' intentions to quit smoking, intentions to use e-cigarettes, and perceptions of relative harm. The study found that misinformation about e-cigarette harms may influence some adult smokers' decisions to consider using e-cigarettes. The manuscript is well-written. However, there are several unclear points in the Methods needed to clarify, as below.

A justification for choosing Twitter (but not other social media sites) as a web-based platform for the experimental should be provided in the Introduction (although it mentioned briefly in the Limitation). If available, provide national percentages of Twitter users in US and UK.	We have now included the following statement in the Introduction: “We used Twitter data because it is free and publicly available and because of the documented prevalence of health misinformation on Twitter.” We have also added the following to the Introduction: “It is estimated that just over one in 5 Americans (22%) and 45% of social media users in the UK use Twitter.”
2. An important point which is not clear is that how to know that participants had been not exposed to the experimented tweets before the study. If they already had seen the assigned tweets before, measures of outcomes may not be the same if they hadn't.	Random assignment to the study's experimental conditions addresses the potential threat of unmeasured confounders such as prior exposure. In addition, participants were asked how often they had seen prior exposure to messages that (1) e-cigarettes are as or more harmful than cigarettes, (2) e-cigarettes are completely harmless. This information is presented in Table 1, indicating equal distribution across the study conditions.
3. Methods - “Randomisation and Masking”: It said that “Participants were then randomised to one of four experimental conditions...”, however, only 3 conditions were listed. In addition, the statement “Adjusting for covariates is thus not needed in subsequent analysis” maybe not correct since adjusting for covariates may be still needed in subsequent	We have now added the fourth condition to the Randomisation and masking section. We have now qualified this statement: “Adjusting for covariates is thus not needed in subsequent analysis, provided randomisation has been successful and covariates are equally distributed across experimental conditions.”

analysis if distributions of confounders are not equal across experimental conditions.	
4. Removing 233 participants who answered 'don't know' to the question on baseline and post-test perceived relative harm of e-cigarettes compared to regular cigarettes out of the analysis may introduce bias if these participants didn't distribute equally across the experimental conditions. I suggest to include this answer option in the analysis.	We have checked and participants who answered 'don't know' to the baseline question regarding relative harm distribute evenly across the experimental conditions, so there is no reason to believe the exclusion of participants who answered don't know from the analysis has resulted in selection bias.
5. Tables 1 and 2 need to provide p-values for testing differences across 4 conditions for the US sample and the UK sample. In addition, name of each condition should be provided rather than the numbers.	We thank the reviewer for raising this critique. We have reviewed the CONSORT guidelines (2010) which state: "significance testing of baseline differences in randomized controlled trials (RCTs) should not be performed, because it is superfluous and can mislead investigators and their readers" (1). Thus, we have opted not to include p-values in Tables 1 and 2. However, if the reviewers and editor disagree with this decision, we would be happy to provide these values. 1. Moher D, Hopewell S, Schulz KF, Montori V, Gøtzsche PC, Devereaux PJ, et al. CONSORT 2010 Explanation and Elaboration: updated guidelines for reporting parallel group randomised trials. J Clin Epidemiol [Internet]. 2010;63(8):e1–37. Available from: http://www.sciencedirect.com/science/article/pii/S0895435610001034 We have now included the name of each condition in addition to their number in Table 1.
6. Tables 3 and 4 can be combined into 1 table	We thank the reviewer for this suggestion, however, when we tried this, we found that combining tables 3 and 4 did not improve their presentation. We therefore reverted back to separate tables in the interests of clearer interpretation for the readers.
7. Why the variable "country" was not included in the main analysis, but only in the sensitive analysis? Was the interaction between "country" and "experimental conditions" examined?	As shown in Table 4 the 95% CI's overlap, we therefore didn't think there would be any interactions. In response to this query, we have now tested for interactions between the country and experimental conditions and found no supporting evidence of their significance. We have added this to the Statistical analysis and Results sections in the manuscript, as follows: "We additionally tested for interactions between experimental conditions and country (US or UK), but found no evidence of an effect."
8. What were the purpose of the sensitivity analyses with robust standard errors and bootstrapping?	The sensitivity analysis using robust standard errors and bootstrapping were undertaken owing to non-normal distribution of residuals. This is stated at the end of the paragraph in the Statistical analysis section.
9. How repeated measures within participants were accounted for in the current analysis?	We only have repeated measures of the outcome variables, which were measured at baseline and post-exposure. We accounted for this by including the baseline measure in the

	adjusted analysis described in paragraph 2 of the Statistical analysis section: “To address the study hypothesis, we utilised linear regression to predict post-test intentions to quit smoking, intentions to purchase e-cigarettes, and perceived relative harm of e-cigarettes by experimental condition compared with the control condition, adjusting for baseline measures of each outcome respectively.” As per your next comment, we have also included them in Tables 3 & 4 to make this clearer.
10. How regression models were adjusted for baseline measures of each outcome? Maybe add these baseline variables in tables for clarification.	We adjusted for baseline measurements of each outcome and have altered Tables 3 & 4 to reflect this.

Reviewer: 2

Dr. Louisa Holmes, The Pennsylvania State University Department of Geography

Comments to the Author:

This study evaluates the impacts of “misinformation” shared via social media on the likelihood of e-cigarette uptake, and perceptions of e-cigarette harm, among an online panel of current cigarette smokers who have never used e-cigarettes in the USA and the UK. The study is sufficiently powered and the comparison regarding e-cigarette and cigarette cessation intent between the USA and the UK, two countries with substantially different policy approaches to e-cigarettes, is innovative.

However, the manuscript has some flaws that should be addressed prior to publication.

ABSTRACT  Line 30, fix so it reads “as or more” rather than “as more” 	Many thanks, we have now changed this.
INTRODUCTION  This manuscript assumes definitions of “misperception” and “misinformation” without ever indicating how those are defined for purposes of the study, or whether they might be subject to bias on the part of the authors (who seem to prefer a harm reduction approach); there are also no citations to support these assumptions so it is difficult to discern whether “misperception” is conflated with “unsupportive of harm reduction.” 	We provided the following definition of ‘misinformation’ in the Introduction: “misinformation —information that is incorrect or misleading.” We have now also provided a definition of misperception as follows: “Misperceptions, defined as false or inaccurate beliefs of the individual,”

METHODS  • In the Methods and Discussion, the authors appear to equate randomization to the study conditions with population distribution, which are not the same (p8, lines 45-50; p13, lines 19-22) – given that recruitment did not reflect population distribution by demographic characteristics, and that e-cigarette use varies substantially along these characteristics (e.g. race/ethnicity, age, socioeconomic position), these items need to be controlled, reported and evaluated in the analyses 	We thank the reviewer for this suggestion. However, successful randomisation addresses the threat of potential confounding by demographic and other characteristics. We have provided details of how randomisation was achieved and evidence of its effectiveness in Table 1 (which shows that the potential confounders are evenly distributed across conditions).
 • P9, lines 5-7: what are the questions that were asked regarding each tweet? 	We have added the following: “After randomisation to a condition, they viewed one tweet at a time in random order (four tweets in total) and were asked brief questions about each tweet -perceived effectiveness of the tweet, likelihood of replying, retweeting, liking and sharing the tweet, and their emotional response to the tweet, more details of these questions can be found in supplemental material 1.”
 • What was the time frame of evaluation from pre- to post-test? 	We have included the following statement to the Procedures section: “The average time taken to complete the survey was 29 minutes.”
 • P9, lines 20-23: how did the team narrow from 499 tweets to 20? What types of criteria were used? 	We describe in the Procedures section how we narrowed this sample of 499 down to 20 tweets: “The study team narrowed this sample of tweets to 20 tweets per experimental condition using the following criteria: 1) explicit statement that e-cigarettes were either as or more harmful than smoking, completely harmless, or uncertain; 2) no mention of children or young people; 3) no mention of specific diseases; 4) no profanities; 5) had multiple ‘likes’ or ‘retweets’; 6) no advertising; 7) no pictures; and 8) was available publicly (i.e., not deleted).”
 • Authors indicate that 50% of the study population did not use Twitter, which is problematic given the study approach; since only half of the population is likely to see Tweets about e-cigarettes, there needs to be more discussion of why this doesn’t skew the study results substantially 	We have added the following to the Discussion: “However, over half of participants (59%) indicated they did not use Twitter meaning they may not be familiar with viewing or engaging with tweets. We included definitions of each of these engagement behaviours, prior to responding to questions on the likelihood of replying, retweeting, liking, or sharing each message.” Further, our findings are still useful because intentions are strong predictors of behaviour, as shown by Ajzen’s Theory of Planned Behavior. Misinformation is ubiquitous - Allcott and colleagues found a total of 672 unique fake news sites. (2) Stories from these sites are shared on Facebook, Twitter and cross-posted on other social media platforms. Therefore, while this sample may not be exposed to misinformation on Twitter in real life, they are likely exposed via different channels.

	2. Allcott H, Gentzkow M, Yu C. Trends in the diffusion of misinformation on social media. Res Polit [Internet]. 2019 Apr 1;6(2):2053168019848554. Available from: https://doi.org/10.1177/2053168019848554
 Literature has demonstrated demographic differences, on average, between Twitter and Facebook users, as well as between those who use social media and those who don't, so authors need to either adjust for this in their models, or list it as a serious limitation (see for example: https://doi.org/10.1177%2F2053168017720008) 	Our sample is not made up of Twitter users (over half of the participants – 59% indicated that they do not use Twitter), rather we have sampled current adult smokers from the US and the UK, who do not currently use e-cigarettes. Further, randomisation addresses any differences in social media use between participants.
 There is no discussion of who the authors of the included Tweets are and how this might impact the way participants (or social media users more generally) interpret the information tweeted, e.g. in Figure 1, the author of the first tweet identifies herself as the founder of an organization, which may confer more authority on her in the participants' mind, as might the fact that Ross Gerber has a blue check, compared with "Brea"  Additionally, there has been much discussion in media, including with Jack Dorsey, about how more likes tends to mean Twitter users are more likely to pay attention to certain Tweets, and there is clear difference in these figures in the number of likes 	We have now included the following in the Discussion section: "Fifth, previous research on health misinformation on social media identified important factors that might play a role in the mechanism of action of misinformation. Among those factors are the type of content, the source of the message, the sender's authoritativeness, the argument length, the novelty, timing, repetition and hashtags. We were not able to examine the impact of these message features in detail. Future research is needed to determine the effects of varying these features on smokers' processing of misinformation about e-cigarettes."
 In Figure 3, the tweet by "Mom folding laundry" is clearly disputing the idea that e-cigarettes are harmful; how can this be classified as "messages expressing uncertainty?" This whole anti-vaping schtick is cooked up by drug regulation & enforcement to make sure the MONEY keeps flowing to their coffers. I have yet to see a single credible piece of evidence that vaping causes real harm. (As in more harm than drinking too much coffee.)	We agree, this tweet is disputing the idea that e-cigarettes are harmful, but it does not say they are completely harmless. Therefore, the reason this is classified as 'uncertain' is because it is not espousing either of our examples of misinformation: (1) e-cigarettes are as or more harmful than regular cigarettes or (2) e-cigarette are completely harmless.
RESULTS  Given the differences in smoking behavior, intent to quit, and e-cig use behavior by sociodemographic characteristic and nationality, it's imperative that regression results for demographic covariates be included 	We thank the reviewer for this suggestion however, it is not necessary to additionally adjust for sociodemographic covariates because we have successfully randomised participants to experimental conditions.

 • Table 1: does White include Non-Hispanic White? In the USA, Latinos have been shown to have substantially different tobacco use patterns than Whites account for nearly 20% of the total population 	In addition to race, we also collected ethnicity as a separate item (non-Hispanic/non-Latinx) among US participants. We have checked and ethnicity is evenly distributed across experimental conditions, so we are confident there is no confounding caused by this. We have updated Table 1 with an additional row showing the distribution of ethnicity among US participants by condition.
LIMITATIONS  • I would suggest that three additional limitations of the study that deserve attention are:  o Lack of inclusion of cannabis use as a covariate given its legality in many parts of the USA and the popularity of using e-cigarettes to vape it o Lack of inclusion of other tobacco products, or polytobacco use, which may confound intentions about e-cigarette use / opinions on e-cig harm o Intention has not been demonstratively linked to behavior among tobacco users 	We did not measure cannabis use because this was outside the scope of this study. This research was funded to address cancer prevention and control specifically. Cannabis use is currently illegal in the UK and many parts of the US. We weighed the potential risks to human subjects of asking this item against the study's objectives and concluded it did not warrant the added risks. Finally, as has been demonstrated, randomisation was achieved which means we are confident that the threat of confounding, but unobserved measures was minimized. We collected data on other tobacco products and poly tobacco use (rolling tobacco, Cigars, Pipe, Hookah, Chew and Snus). Again, we have checked and these measures were evenly distributed across experimental conditions, so we are confident there is no confounding caused by this. While intention to do something is not the same as doing it, there is substantial evidence that smoking behaviour is related to smoking intentions. Consistent with the Theory of Planned Behaviour hypotheses, a systematic review reported that smoking behaviour was related to intentions (mean weighted effect size $r=0.30$ across 35 datasets). (3) 3. Topa G, Moriano JA. Theory of planned behavior and smoking: meta-analysis and SEM model. Subst Abuse Rehabil [Internet]. 2010 Dec 6;1:23–33. Available from: https://pubmed.ncbi.nlm.nih.gov/24474850 However, in response to this comment we have added the following to the end of the Discussion section: “Finally, future research could extend our analysis to include behaviours as well as intentions.”
DISCUSSION  • I would respectfully suggest that the differences in results by country are one of the more interesting aspects of the study and that these should be included in the discussion with attention to how cultural and policy approaches to e-cigarettes clearly seem to have an impact on intentions and harm perception 	We have now added the following to the discussion: “We also observed that US smokers who viewed tweets that e-cigs were completely harmless reported lower perceived harms of vaping and higher intentions to purchase e-cigarettes in this study. This effect was absent among UK smokers. This difference between US and UK smokers may be due to the differing policy contexts of the countries. However, further research is needed to assess underlying

	policy and contextual factors that explain these differences between countries in the effects of e-cigarette misinformation.”
--	---

References

1. Moher D, Hopewell S, Schulz KF, Montori V, Gøtzsche PC, Devereaux PJ, et al. CONSORT 2010 Explanation and Elaboration: updated guidelines for reporting parallel group randomised trials. *J Clin Epidemiol* [Internet]. 2010;63(8):e1–37. Available from: <http://www.sciencedirect.com/science/article/pii/S0895435610001034>
2. Allcott H, Gentzkow M, Yu C. Trends in the diffusion of misinformation on social media. *Res Polit* [Internet]. 2019 Apr 1;6(2):2053168019848554. Available from: <https://doi.org/10.1177/2053168019848554>
3. Topa G, Moriano JA. Theory of planned behavior and smoking: meta-analysis and SEM model. *Subst Abuse Rehabil* [Internet]. 2010 Dec 6;1:23–33. Available from: <https://pubmed.ncbi.nlm.nih.gov/24474850>

VERSION 2 – REVIEW

REVIEWER	Nguyen, Nhung UCSF
REVIEW RETURNED	01-Mar-2021

GENERAL COMMENTS	I appreciate the authors’ efforts to address the concerns raised by the reviewers and to improve the previous manuscript. However, there are still issues to be addressed as below.  1. "Participants who answered ‘don’t know’ to the baseline question regarding relative harm distribute evenly across the experimental conditions and therefore pose no problem with respect to confounding or selection bias."  How about participants who answered “don’t know” at post-test? Did they distribute equally? What the planned analysis for this randomised trial? Intent-to-treat or else? 2. In the Statistical Analysis: “To address the study hypothesis, we utilised linear regression to predict ...”  What is the study hypothesis? The hypothesis needs to be provided in the Introduction. 3. It is important to know % of Twitter users across 4 conditions. These estimates should be provided in Table 1 4. In the Results, “The adjusted analysis includes both the experimental condition as the exposure and the baseline measure of the outcomes.”  Assuming randomization was successful, the baseline measures of outcomes would be distributed evenly across 4 conditions, so including these measures in the regression is not necessary. In addition, inclusion of the baseline measures of the outcome did not control for correlated measures of pre- and post- for an outcome within participants. Another better approach for repeated measure analysis (e.g., GLM) should be considered. 5. In the Results “We additionally tested for interactions between experimental conditions and country (US or UK), but found no evidence of an effect”  If so, there is no need to report the results of stratified analysis by country (Table 4), and there is not relevant to discuss any differences between two countries in the Discussion
---

	6. It is hard to judge whether the twists about “e-cigarettes are equally or more harmful than smoking regular cigarettes” is truly misperception/misinformation or not, since the extant evidence on long-term harms of e-cigarette is limited and the outbreak of lung injury related to vaping in the US in 2019 raised a concern about emerging deleterious health effects of e-cigarette use. Given the study results in Table 3, I am confused about suggestions to correct misperceptions. Which misperceptions should be corrected and how to correct them? 7. The discussion on “misinformation about e-cigarettes may be hindering efforts to reduce the burden of tobacco smoking on current smokers” is not supported by the study results since no significant associations were found for intention to quit smoking cigarettes. I also suggest to revise the whole discussion to not overstate the results. 8. The Discussion should discuss about effects of a quit short time (~30 min) of exposure to the twists in this study. This very brief exposure might not be enough to make any significant changes in the outcomes. 9. Other: This manuscript should be shortened. There is an error in the abstract: ($\beta=-0.154, 95\%CI:-0.258, 0.050$) should be $95\%CI: -0.258, -0.050$.
--	---

REVIEWER	Holmes, Louisa The Pennsylvania State University Department of Geography, GEOG
REVIEW RETURNED	25-Feb-2021

GENERAL COMMENTS	The authors have made considerable improvement to the paper, and I thank them for attention to reviews. However, I still dispute the idea that population characteristics have no bearing because the study was randomized. Randomization without stratifying by individual characteristics or behaviors does not ensure equal distribution to study conditions despite the authors' insistence. Randomization is not the same thing as ensuring equitable population distribution across cases, and given the substantial differences in use of e-cigarettes and social media across different individual strata, this needs to be addressed in the analysis somehow, or at the very least included in the limitations. Otherwise the additions/edits to the manuscript are appropriate.
--

VERSION 2 – AUTHOR RESPONSE

Reviewer: 2

Dr. Louisa Holmes, The Pennsylvania State University Department of Geography

Comments to the Author:

The authors have made considerable improvement to the paper, and I thank them for attention to reviews. However, I still dispute the idea that population characteristics have no bearing because the study was randomized. Randomization without stratifying by individual	We thank the reviewer for this comment. We believe this query refers to stratified randomization, which aims to ensure equal proportions of participants are in the treatment and control arms of an experiment based on various characteristics. We refer the reviewer to this JCE review: https://pubmed.ncbi.nlm.nih.gov/9973070/ (1). This review shows that stratified randomization is more typically used in small clinical trials (n≤400) where covariates with large effects on the outcomes are included. We believe stratified
--	---

characteristics or behaviors does not ensure equal distribution to study conditions despite the authors' insistence. Randomization is not the same thing as ensuring equitable population distribution across cases, and given the substantial differences in use of e-cigarettes and social media across different individual strata, this needs to be addressed in the analysis somehow, or at the very least included in the limitations. Otherwise the additions/edits to the manuscript are appropriate.	randomization is unnecessary for the following reasons: (1) this is not a small trial (n=2,400), (2) we had no a priori hypotheses about population-level variables known to have large effects on our outcome and (3) we have successfully randomized (see Table 1, which shows even distributions of these population-level measures, across the 4 arms of the experiment), which means no adjustment is required. To demonstrate this assertion, we include below a table which compares the (unadjusted) analysis which was included in the previously reviewed manuscript (in Table 3) and a fully adjusted analysis, which includes a number of covariates - country, sex, education, income, race, age, previous e-cigarette use, internet use and twitter use. As shown in the Table, the inclusion of this extensive list of covariates makes little or no difference to the estimates, and no difference to the conclusions drawn.
--	--

(1) Kernan WN, Viscoli CM, Makuch RW, Brass LM, Horwitz RI. Stratified randomization for clinical trials. *J Clin Epidemiol.* 1999 Jan;52(1):19-26. doi: 10.1016/s0895-4356(98)00138-3. PMID: 9973070.

Table 1: comparing the unadjusted analysis presented in previously reviewed manuscript, with a fully adjusted analysis as suggested by reviewer 2

	Unadjusted (results presented in original paper)			Adjusted for population-level measures (country, sex, education, income, race, age, previous e-cigarette use, internet use & twitter use) in response to reviewer 2 comments		
	β	95% CI	p	β	95% CI	p
Intention to quit						
US						
Control (referent)						
As or more harmful	-0.13	-0.31,0.05	0.169	-0.13	-0.31,0.05	0.167
Completely harmless	-0.16	-0.34,0.02	0.079	-0.16	-0.34,0.02	0.077
Uncertainty	-0.02	-0.20,0.15	0.786	-0.03	-0.21,0.15	0.761
Pre-exposure intention to quit	0.94	0.92,0.96	≤ 0.001	0.94	0.92,0.96	≤ 0.001
UK						
Control (referent)						
As or more harmful	0.06	-0.10,0.23	0.451	0.06	-0.10,0.23	0.457
Completely harmless	-0.08	-0.24,0.09	0.344	-0.08	-0.24,0.09	0.351
Uncertainty	-0.01	-0.18,0.15	0.895	0.001	-0.16,0.16	0.988
Pre-exposure intention to quit	0.95	0.93,0.97	≤ 0.001	0.94	0.92,0.96	≤ 0.001
Intention to purchase						
US						
	β	95% CI	p	β	95% CI	p

Control (referent)						
As or more harmful	-0.33	-0.52,-0.14	≤0.001	-0.32	-0.51,-0.13	0.001
Completely harmless	0.15	-0.05,0.34	0.137	0.15	-0.04,0.34	0.133
Uncertainty	-0.08	-0.27,0.11	0.401	-0.08	-0.27,0.11	0.434
Pre-exposure intention to purchase	0.84	0.81,0.87	≤0.001	0.83	0.80,0.86	≤0.001
UK						
Control (referent)						
As or more harmful	-0.55	-0.75,-0.34	≤0.001	-0.54	-0.75,0.33	≤0.001
Completely harmless	0.01	-0.19,0.22	0.893	0.02	-0.18,0.23	0.816
Uncertainty	-0.20	-0.41,0.003	0.054	-0.20	-0.40,0.01	0.065
Pre-exposure intention to purchase	0.86	0.83,0.89	≤0.001	0.86	0.83,0.89	≤0.001
Relative harm						
US	β	95% CI	p	β	95% CI	p
Control (referent)						
As or more harmful	0.30	0.19,0.40	≤0.001	0.29	0.19,0.40	≤0.001
Completely harmless	-0.15	-0.26,-0.05	0.05	-0.16	-0.26,-0.05	0.003
Uncertainty	0.04	-0.07,0.14	0.492	0.03	-0.08,0.13	0.588
Pre-exposure relative harm	0.81	0.77,0.84	≤0.001	0.80	-0.77,0.84	≤0.001
UK						
Control (referent)						
As or more harmful	0.39	0.30,0.47	≤0.001	0.39	0.30,0.48	≤0.001
Completely harmless	-0.05	-0.14,0.04	0.238	-0.05	-0.14,0.03	0.228
Uncertainty	-0.002	-0.09,0.09	0.958	-0.007	-0.10,0.08	0.879
Pre-exposure relative harm	0.87	0.84,0.91	≤0.001	0.86	0.82,0.89	≤0.001

Reviewer: 1

Dr. Nhung Nguyen, UCSF

Comments to the Author:

I appreciate the authors' efforts to address the concerns raised by the reviewers and to improve the previous manuscript. However, there are still issues to be addressed as below.

1. "Participants who answered 'don't know' to the baseline question regarding relative harm distribute evenly across the experimental conditions and therefore pose no problem with respect to confounding or selection bias."  How about participants who answered "don't know" at post-test? Did	Apologies this was not made clearer. Those who answered 'don't know' either at baseline or post-exposure, were equally distributed across the experimental conditions and we have edited the manuscript to reflect this. The CONSORT diagram (Figure 5) reflects
---	---

they distribute equally? What the planned analysis for this randomised trial? Intent-to-treat or else?	those who answered don't know either at baseline or post-exposure. This is not an intention to treat analysis, rather we are following our protocol for the analysis, which is a complete case analysis i.e. all participants who did not have any missing data.
2. In the Statistical Analysis: "To address the study hypothesis, we utilised linear regression to predict ..."  What is the study hypothesis? The hypothesis needs to be provided in the Introduction.	We agree this is unclear and have removed the word hypothesis and replaced it with the term study aims.
3. It is important to know % of Twitter users across 4 conditions. These estimates should be provided in Table 1	We agree, we have now included this in Table 1.
4. In the Results, "The adjusted analysis includes both the experimental condition as the exposure and the baseline measure of the outcomes."  Assuming randomization was successful, the baseline measures of outcomes would be distributed evenly across 4 conditions, so including these measures in the regression is not necessary. In addition, inclusion of the baseline measures of the outcome did not control for correlated measures of pre- and post- for an outcome within participants. Another better approach for repeated measure analysis (e.g., GLM) should be considered.	We agree with you; and we included these coefficients in Table 3 at the request of the other reviewer. We will leave it up to the editor to decide whether these should be included in the final manuscript. Thank you for your suggestion, given the design of our study (a single exposure, over 30 minutes) we believe our method is a good fit. If we were to collect additional data points we would adopt a repeated measures approach, as alluded to in the final paragraph of the Discussion.
5. In the Results "We additionally tested for interactions between experimental conditions and country (US or UK), but found no evidence of an effect"  If so, there is no need to report the results of stratified analysis by country (Table 4), and there is not relevant to discuss any differences between two countries in the Discussion	Thank you for this comment, but we would respectfully disagree. Only reporting 'significant' results leads to outcome reporting bias. This analysis was included as it was one of the study aims to explore the different contexts of the US and the UK. We believe it is relevant and of interest to keep the comparisons in the manuscript.
6. It is hard to judge whether the twists about "e-cigarettes are equally or more harmful than smoking regular cigarettes" is truly misperception/misinformation or not, since the extant evidence on long-term harms of e-cigarette is limited and the outbreak of lung injury related to vaping in the US in 2019 raised a concern about emerging deleterious health effects of e-cigarette use. Given the study results in Table 3, I am confused about suggestions to correct misperceptions. Which misperceptions should be corrected and how to correct them?	Many thanks for this comment. As was mentioned in the paper, we based the definitions of misinformation on the current state of the science of e-cigarette harms, (1,2) tweets that e-cigarettes are just as or more harmful than smoking are defined as misinformation based on the known short-term harms. We define misinformation as follows: "misinformation related to e-cigarette harms was classified as the statements that either claim that e-cigarettes are equally or more harmful than smoking regular cigarettes or are completely harmless." "Misperceptions, defined as false or inaccurate beliefs of the individual" Therefore, if smokers believe that e-cigarettes are just as harmful as cigarettes, or

	completely harmless, these are misperceptions (based on the current evidence). We addressed the occurrence of the EVALI outbreak prior to the data collection period within the discussion. It is also important to note that the EVALI outbreak has been linked to vaping THC liquids and Vitamin E acetate and not nicotine-based e-cigarettes, which is the primary focus of this research. Interventions targeting/correcting misperceptions are beyond the scope of this research. We do however, suggest that future research could focus on this in the Discussion.
7. The discussion on “misinformation about e-cigarettes may be hindering efforts to reduce the burden of tobacco smoking on current smokers” is not supported by the study results since no significant associations were found for intention to quit smoking cigarettes. I also suggest to revise the whole discussion to not overstate the results.	Thank you for flagging this up, we have removed this sentence and changed the text as follows: “These findings are important because they show that after brief exposure to tweets that e-cigarettes are as or more harmful than smoking, current smokers may be deterred from using e-cigarettes (measured with intention to purchase e-cigarettes) as a harm reduction strategy. They are also more likely to wrongly believe that e-cigarettes are more harmful than regular cigarettes. However, more research is needed to assess whether misinformation exposure about e-cigarette harms will negatively influence smokers' behaviours to reduce harms from using combusted cigarettes by opting for less harmful forms of nicotine delivery using e-cigarettes.” We have also updated the Strengths and limitations of this study section to reflect this change.
8. The Discussion should discuss about effects of a quit short time (~30 min) of exposure to the twists in this study. This very brief exposure might not be enough to make any significant changes in the outcomes.	We thank the reviewer for this comment, and direct them to the final paragraph of the Discussion where we refer to this, we have added the text to further clarify this point: “Our exposure was only brief therefore, future research to evaluate the effect of longer or repeated exposures to misinformation would also be useful,”
9. Other: This manuscript should be shortened. There is an error in the abstract: ($\beta = -0.154, 95\%CI: -0.258, 0.050$) should be $95\%CI: -0.258, -0.050$.	We confirm that our article does not exceed the journal's word limit of 4000 words. Thank you for highlighting this error, which has now been amended in the manuscript.